# Two-Stage Deep Learning Model for Automated Segmentation and Classification of Splenomegaly

**DOI:** 10.3390/cancers14225476

**Published:** 2022-11-08

**Authors:** Aymen Meddeb, Tabea Kossen, Keno K. Bressem, Noah Molinski, Bernd Hamm, Sebastian N. Nagel

**Affiliations:** 1Charité—Universitätsmedizin Berlin, Corporate Member of Freie Universität Berlin and Humboldt-Universität zu Berlin, Klinik für Radiologie, Hindenburgdamm 30, 12203 Berlin, Germany; 2CLAIM—Charité Lab for AI in Medicine, Charité—Universitätsmedizin Berlin, Augustenburger Platz 1, 13353 Berlin, Germany; 3Berlin Institute of Health, Charité—Universitätsmedizin Berlin, Charitéplatz 1, 10117 Berlin, Germany; 4Charité—Universitätsmedizin Berlin, Corporate Member of Freie Universität Berlin and Humboldt-Universität zu Berlin, Institut für Neuroradiologie, Charitéplatz 1, 10117 Berlin, Germany

**Keywords:** malignant lymphoma, splenic involvement, radiomics, machine learning, computer aided diagnosis, subtype classification, quantitative imaging biomarkers

## Abstract

**Simple Summary:**

Splenomegaly is a feature of a broad range of diseases including hematological malignancies and non-neoplastic conditions. However, the morphological appearance of an enlarged spleen alone does not necessarily reveal the underlying cause. The application of deep learning could deliver new quantitative imaging biomarkers to identify the underlying etiology of splenomegaly. In this study, a deep learning model was developed to automatically segment and classify splenomegaly in patients with malignant lymphoma versus patients with cirrhotic portal hypertension based on CT images. This model could help identify the underlying disease and triaging malignant cases to ensure timely diagnosis and treatment.

**Abstract:**

Splenomegaly is a common cross-sectional imaging finding with a variety of differential diagnoses. This study aimed to evaluate whether a deep learning model could automatically segment the spleen and identify the cause of splenomegaly in patients with cirrhotic portal hypertension versus patients with lymphoma disease. This retrospective study included 149 patients with splenomegaly on computed tomography (CT) images (77 patients with cirrhotic portal hypertension, 72 patients with lymphoma) who underwent a CT scan between October 2020 and July 2021. The dataset was divided into a training (*n* = 99), a validation (*n* = 25) and a test cohort (*n* = 25). In the first stage, the spleen was automatically segmented using a modified U-Net architecture. In the second stage, the CT images were classified into two groups using a 3D DenseNet to discriminate between the causes of splenomegaly, first using the whole abdominal CT, and second using only the spleen segmentation mask. The classification performances were evaluated using the area under the receiver operating characteristic curve (AUC), accuracy (ACC), sensitivity (SEN), and specificity (SPE). Occlusion sensitivity maps were applied to the whole abdominal CT images, to illustrate which regions were important for the prediction. When trained on the whole abdominal CT volume, the DenseNet was able to differentiate between the lymphoma and liver cirrhosis in the test cohort with an AUC of 0.88 and an ACC of 0.88. When the model was trained on the spleen segmentation mask, the performance decreased (AUC = 0.81, ACC = 0.76). Our model was able to accurately segment splenomegaly and recognize the underlying cause. Training on whole abdomen scans outperformed training using the segmentation mask. Nonetheless, considering the performance, a broader and more general application to differentiate other causes for splenomegaly is also conceivable.

## 1. Introduction

As a major lymphoid organ, the spleen is involved in a broad range of diseases, including hematological malignancies as well as infectious or inflammatory syndromes [1,2,3]. Splenic infiltration can lead to changes in volume, morphology, and metabolic activity [4]. The discrimination of these changes based on qualitative features of cross-sectional imaging alone can be challenging [4], and often only a rather descriptive reference is reported. Although splenomegaly is a common finding in computed tomography (CT), often only the craniocaudal diameter is mentioned in radiological reports. An accurate segmentation and classification of splenomegaly could help identify the underlying disease and triaging malignant cases to ensure timely diagnosis and treatment, but manual segmentation is time consuming and not feasible in clinical routine.

In the case of malignant hematological diseases, besides the classification according to the World Health Organization’s (WHO) classification of tumors of hematopoietic and lymphoid tissues [5], cross-sectional imaging plays a significant role in baseline staging and therapy monitoring [6]. Splenomegaly is seen in approximately one third of all Hodgkin lymphomas (HL) at presentation [7] and in different proportions in other subtypes [8]. Previous retrospective studies in US hospitals reported that hematological diseases were found in 16–66% in patients with splenomegaly: among them, the most common diagnoses were lymphoma (16–44% of all splenomegaly), CML (8–29%), hemoglobinopathy (7–25%), CLL (0–20%), and myelofibrosis (9–16%) [9,10,11]. In a baseline imaging workup, splenomegaly is defined by more than 13 cm of craniocaudal diameter in CT images according to the Lugano classification of lymphomas [12]. 

In recent years, deep learning achieved high performance in segmentation and classification tasks in medical imaging, with some algorithms being successfully implemented in clinical routine [13,14,15]. In oncologic imaging, the accurate and automated segmentation of abdominal organs is a critical first step for the detection and delineation of tumors and metastases, and for surgical preplanning. Recent works showed robust results in the spleen, kidneys and liver segmentation on CT and MRI images with high dice scores ranging from 0.88 to 0.96 [16,17,18,19,20]. Other works demonstrated the value of imaging biomarkers in differentiating malignant lymphoma from other cancer entities [21,22], and even in predicting early relapse, as it has been shown by Lisson et al. for mantle cell lymphoma [23].

In our study, we present a two-stage deep learning model to automatically segment the spleen and distinguish whether splenomegaly originates from lymphoma or from cirrhotic portal hypertension. Such a model would provide a highly accurate segmentation of the spleen, and contribute to the effort of developing an imaging biomarker to analyze splenic changes in oncological and cirrhotic patients. To our knowledge, this is the first study to evaluate an automated, deep learning-based segmentation, and classification of causes of splenomegaly. 

## 2. Materials and Methods

This retrospective study was approved by our institutional review board (No.: EA4/136/21). The requirement for informed consent was waived due to the retrospective design of the study. The personal data of the patients were strictly protected and anonymized prior to analysis. 

### 2.1. Study Population

The inclusion criteria were either (1) patients diagnosed with liver cirrhosis or (2) patients with lymphoma. 

The exclusion criteria were (1) normal splenic volume; (2) splenic infarction; (3) cystic or solid lesions of the spleen; and (4) thrombosis of the portal or the splenic vein.

Enrolled patients were randomly divided into a training cohort (*n* = 124) and a testing cohort (*n* = 25). The study flow diagram is shown in Figure 1.

### 2.2. CT Imaging Characteristics and Scanning Protocol

For the lymphoma patients, the baseline staging protocol was standardized and included a contrast-enhanced CT of the neck, chest, abdomen and pelvis. In patients with liver cirrhosis, the study protocol included either the chest, abdomen, and pelvis, or only abdomen and pelvis, depending on the clinical question. 

CT scanners from two manufacturers were used to acquire the CT scans: Aquilion One (number of performed examinations = 22), Aquilion PRIME (*n* = 38), and Aquilion 64 (*n* = 2) from Canon Medical Systems (Otawara, Tochigi, Japan) and Revolution HD (*n* = 37), Revolution EVO (*n* = 42) and LightSpeed VCT (*n* = 8) from General Electric Healthcare (Boston, MA, USA). 

The contrast agents used were iomeprol (Imeron 400^®^, Bracco Imaging, Milan, Italy) iobitridol (Xenetix 350^®^, Guerbert, Villepinte, France), and iopromide (Ultravist 370^®^, Bayer, Leverkusen, Germany) with amounts varying between 100 and 140 mL. Portal venous phase imaging was performed at 70–80 s after the intravenous administration of the contrast agent. Axial reconstructions with a slice thickness of 5 mm without gaps were used in this study. Figure 2 shows a sample of CT images in coronal reconstruction.

### 2.3. Segmentation Pipeline

The deep learning-based model was developed on the open source MONAI Framework (Medical Open Network for AI, version 0.8.0) [24]. The automated segmentation of the spleen in CT images was performed using a 3D U-Net architecture and already presented in our previous study [13]. Briefly, the model consisted of an enhanced version of 3D U-Net with residual units, which was trained on an open dataset and an inhouse dataset (a total of 122 patients). A 3D U-Net consists of contracting an expansive path (downsampling and upsampling). In our study, five downsampling and upsampling blocks were implemented. The downsampling block consisted of two convolutional layers (convolution—instance normalization—parametric rectified linear unit (PReLU)) and a skip-connection representing a residual unit. During training, Dice loss as loss function and Adam as optimizer were used, with a learning rate set at 1e-4. The implemented 3D U-Net achieved a dice score of 0.941 ± 0.021. The cohort presented in this study was not included in the training of the segmentation model.

### 2.4. Classification Pipeline

The image data were reformatted from standard DICOM to the Neuroimaging Informatics Technology Initiative (NIfTI) format and subsequently transferred to an in-house server for the training, validation, and testing of the model. 

The 3D DenseNet was implemented using the Python programming language (version 3.7, Python Software Foundation, https://www.python.org (accessed on 17 April 2022)) on the open source deep learning framework MONAI in conjunction with PyTorch (version 1.8.1 https://pytorch.org (accessed on 17 April 2022)).

A DenseNet is a convolutional neural network (CNN) which is composed of four dense blocks as represented in the schematic diagram (Figure 3), and which connects each layer to every other layer in a feed-forward fashion [25]. For our study, we implemented the DenseNet model due to various advantages: first, it combines features by concatenating them into all subsequent layers. Each layer gains the collective knowledge of all other layers, resulting in a thinner and compact network. Second, DenseNet 121 has fewer trainable parameters and is less computationally intensive compared to ResNet 50 [25]. To investigate whether the use of DenseNet 121 is beneficial, another CNN with residual connections (ResNet 50) was additionally evaluated.

During training, we used the Dice loss as the loss function and Adam as the optimizer, with a learning rate set at 1e-4 and backpropagation to compute the gradient of the loss function.

## 3. Results

### 3.1. Study Population

A total of 149 patients were enrolled in this study, as shown in Figure 1, among which 77 patients had liver cirrhosis with splenomegaly due to portal hypertension, and 72 patients had a histologically proven malignant hematologic disease, including 12 different subtypes—namely diffuse large B-cell lymphoma (DLBCL), follicular lymphoma (FL), T-cell lymphoma (TCL), Burkitt lymphoma (BL), Hodgkin lymphoma (HL), mantle cell lymphoma (MCL), acute myeloid leukemia (AML), acute lymphoblastic leukemia (ALL), marginal zone lymphoma (MZL), chronic myeloid leukemia (CML), plasmablastic lymphoma (PBL), and post-transplant lymphoproliferative disorder (PTLD). CT images were acquired between October 2020 and July 2021. Patient and disease characteristics are outlined in Table 1. Since more than 60% of the patients presented with a diffuse large B-cell lymphoma (DLBCL), a follicular lymphoma (FL), and B-cell lymphoma (BCL), only these three subtypes are included in Table 1. All the remaining subtypes are among others**. 

### 3.2. Classification Performance Using the Spleen Mask

When trained only with the spleen mask, DenseNet achieved an ACC of 0.84 and an AUC of 0.84 in the validation dataset, and an ACC of 0.76 and an AUC of 0.81 in the test set. ResNet achieved an ACC of 0.79 and an AUC of 0.82 in the validation, and an ACC of 0.64 and an AUC of 0.77 in the testing cohort. The metrics of DenseNet are summarized in the confusion matrix and the ROC AUC curve (Figure 4) and in Table 2.

### 3.3. Classification Performance Using the Whole Abdominal Volume (Dense-Abd)

The implemented 3D DenseNet for distinguishing splenomegaly in patients with malignant lymphoma patients from those with liver cirrhosis achieved an ACC of 0.88 and an AUC of 0.88 in the validation and the testing cohort. ResNet achieved an ACC of 0.84 and an AUC of 0.86 in the validation, and an ACC of 0.80 and an AUC of 0.80 in the testing cohort. The metrics for DenseNet are summarized in the confusion matrix and the ROC AUC curve (Figure 5) and in Table 2.

### 3.4. Occlusion Sensitivity Maps Visualization

In order to understand the decision making of our model, we analyzed the occlusion sensitivity maps of the whole abdominal volumes. The occlusion map shows which parts of the image have a positive contribution and which parts have a negative contribution to the classification score. Red areas of the map have higher and blue areas lower values, where higher values represent parts of the image that lead to a decrease of the score when occluded, i.e., they contain important information. Three regions were mostly involved in the prediction of the underlying disease, namely the liver, the spleen, and the periaortic retroperitoneum. Figure 6 shows exemplary CT images and the corresponding occlusion sensitivity maps superimposed over the input images. The visualization reveals that the deep learning model makes the classification of whether the patient has a splenomegaly due to liver cirrhosis with portal hypertension or due to a malignant lymphoma by focusing not only on the spleen, but also on the liver and to a lesser extent to retroperitoneal lymphadenopathy. 

## 4. Discussion

The aim of this study was to develop a two-stage deep learning-based model for automated segmentation and classification of causes of splenomegaly. The results show that (i) an automated segmentation of the spleen can be applied to detect and delineate splenomegaly; (ii) a deep learning classification model can accurately differentiate between splenomegaly due to a malignant lymphoma and due to liver cirrhosis with portal hypertension; and (iii) when only trained with the spleen mask, the classification model still shows a high accuracy in discriminating the causes of splenomegaly. 

In recent years, deep learning has gained momentum in different computer vision fields and is more and more applied for segmentation and classification tasks in medical image analysis. Deep learning-based segmentation models even reached the human level in detecting and delineating organs and pathologies [14,15,27,28,29]. To the best of our knowledge, this is the first study to combine a deep learning pipeline for spleen segmentation and classification of the underlying disease of splenomegaly.

As a major lymphoid organ, the spleen is involved in a wide range of infectious, metabolic and hematological diseases, and splenomegaly is a frequent image finding. Whereas splenomegaly is present in up to two thirds of patients with hematological conditions [9], it has been described in up to 50% of patients with liver cirrhosis and portal hypertension [30]. The spleen parenchyma is divided into two distinct macroscopic compartments, the red pulp and the white pulp. The red pulp is composed of venous sinuses, reticular fibers, myofibroblasts, and associated macrophages. The white pulp is composed of the peri-arteriolar lymphoid sheath (PALS), the follicles, and the marginal zone, which includes lymphocytes, macrophages, dendritic cells, and plasma cells [31]. Etiologically, liver cirrhosis and portal hypertension results in a congestive splenomegaly, which is characterized by a prominent red pulp, whereas hematological disorders induce an infiltration of the white pulp and marginal zone by clonal hematopoietic cells [30]. Subsequently, the different histological patterns could result in different imaging features. In our study, we evaluated whether a deep learning-based classification model could discriminate between the two etiologies of splenomegaly using CT images.

Because of different histopathological characteristics, several studies have also explored the usefulness of radiomics signatures to further characterize conditions with splenic involvement. Radiomics is a machine learning method that extracts and analyzes quantitative imaging features and textural information to enhance clinical-decision making [32]. A recent study, for example, showed that splenic radiomics features on CT can predict the prognosis of gastric cancer patients [33]. Another study used the spleen’s radiomics signature on CT to predict the recurrence of HCC [34]. Additionally, considering lymphoma, Enke et al. developed a radiomics model on CT to differentiate between spleen involvement versus controls, as well as to discriminate different subtypes of malignant lymphoma [35]. Their results showed that the radiomics signature could predict the presence of malignant lymphoma with an AUC of 0.86, and even differentiate between subtypes with a satisfying AUC. However, in spite of promising results, the main challenge of radiomic studies is their poor reproducibility, since published data suggest that all steps prior to radiomics analysis can bias feature values [36,37]. Even though they were not explicitly considered in our study, the two-stage approach suggests that textural features may play a role in determining the cause of splenomegaly, since the model could only consider the spleen to make its decision.

In the first step of our study, splenomegaly was automatically segmented with a 3D U-Net model, reaching a dice score of 0.94, which is comparable with the results of Humpire-Mamani et al. [13,19]. More recently, modified U-Net architectures, such as Attention-based U-Net, TransUNet, and Swin-UNet achieved higher scores in medical imaging segmentation, as demonstrated by Gulzar et al. [38], but these models are more complex than the U-Net and necessitate more computational power. The automated segmentation of splenomegaly can also save precious time, eliminate interrater variability, and allow quantitative imaging analysis with a higher amount of data and its implementation in clinical routine. In the second step, 3D DenseNet and 3D ResNet models were trained and validated once on the whole abdominal volume, and once only using the spleen masks, automatically generated from the 3D U-Net model in step one. Using the whole abdominal CT-image, the DenseNet model reached an AUC of 0.88 and an ACC of 0.88. When the spleen mask is applied, the robustness of the model slightly drops, but still shows a satisfactory AUC of 0.81 and an ACC of 0.76. ResNet showed a slightly lower performance as described in Table 2. Since disease classification in medical imaging is a relatively new field, there is still no global consensus on which the model would perform the best. Luetkens et al. used ResNet-50 and DenseNet-121 for the differentiation of alcoholic and other-than-alcoholic cirrhosis based on MRI. ResNet50 achieved the best results (ACC 0.75, AUC 0.82), however, the performance was not significantly higher compared to Densenet121 [39]. Remedios et al. provided an ablation study to compare convolutional neural networks for detecting large-vessel occlusion on computed tomography angiography in 300 patients. The performances of ResNet-50, DenseNet-121, EfficientNet-B0, PhiNet, and an Inception module-based network were compared. An external validation set showed that DenseNet-121 had the best average performance on accuracy, precision, recall, specificity, and F1 score. In concordance with these studies, DenseNet-121 performed better than ResNet for our classification task. 

To better understand the decision-making process of our classification model, we applied the occlusion sensitivity maps to the abdominal CT-image. We identified three regions of interests that were frequently enhanced on the heat maps, namely the liver, the spleen, and the periaortic retroperitoneum. First, this observation demonstrated that our classification model recognized the important imaging features and spared other parts of the abdomen that were not relevant to the classification task. Moreover, this reveals the presence of relevant imaging features in the liver, in the spleen, and in the retroperitoneum to differentiate between splenomegaly due to malignant lymphoma and due to liver cirrhosis with portal hypertension.

As an outlook, our deep learning-based pipeline could be easily integrated into the clinical workflow in a triage scenario [40]. After image acquisition, the deep learning model can automatically segment the spleen, quantify its volume, and predict the underlying disease. The patients could even be automatically referred to the right physician (hematologist for malignant lymphoma, gastroenterologist for liver cirrhosis).

Our study has some potential limitations. First, as a retrospective, single-center study, an external validation of the model robustness is missing. Moreover, our cohort with 149 patients is relatively small. However, our deep learning model cannot be considered as a “black box” since the occlusion sensitivity maps provide a reasonable explanation of the classification outcomes. Since we focused on a dichotomous decision regarding splenomegaly and its causes in our study, we cannot generalize it to all patients with splenomegaly. Splenomegaly also has a limited value in determining splenic involvement in malignant lymphomas, as one-third of normal-sized spleens can have a focal or diffuse tumor infiltration without splenomegaly [41]. Therefore, we will include all patients with diagnosed malignant lymphoma in the future, with new features such as splenic lesions, and evaluate a classification model to differentiate between the subtypes of malignant hematological diseases. 

## 5. Conclusions

We present a deep learning-based pipeline for a fully automated segmentation of splenomegaly and classification of its etiology. Our model can attain a state-of-the-art performance in detecting and segmenting splenomegaly, and the binary classification is capable of differentiating between malignant lymphoma and liver cirrhosis as the underlying disease. Hence, considering the model performance, a broader and more general application to differentiate other causes for splenomegaly is also conceivable.

## Figures and Tables

**Figure 1 cancers-14-05476-f001:**
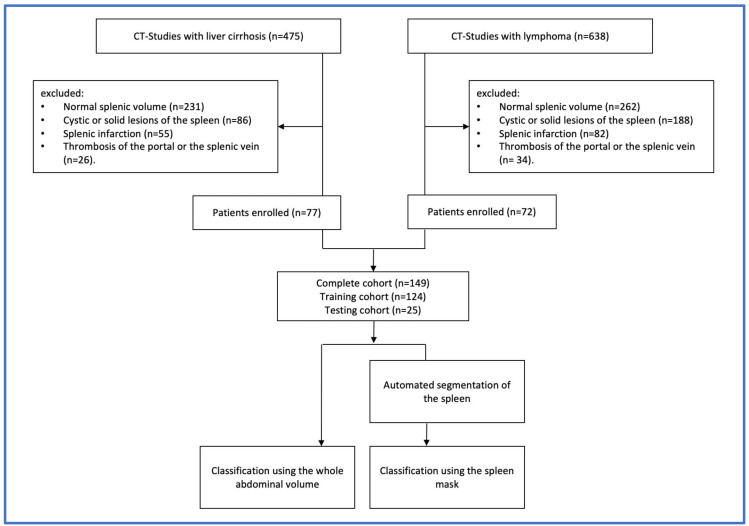
Study flow diagram showing the steps conducted for the selection of samples and the study design.

**Figure 2 cancers-14-05476-f002:**
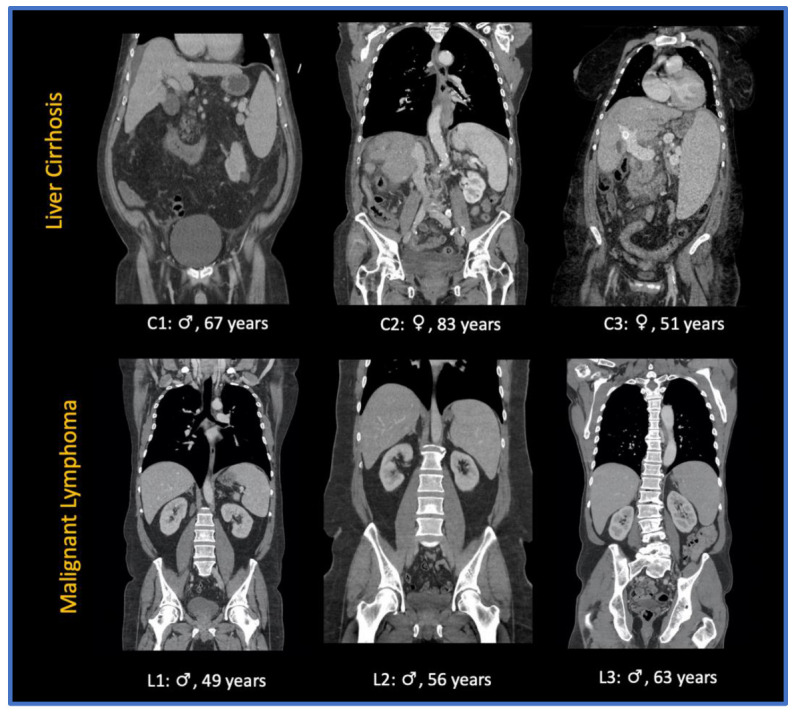
Sample of CT images of patients presenting with splenomegaly due to liver cirrhosis (C1-3) and splenomegaly due to malignant lymphoma (L1-3): ♀: female patient; ♂: male patient.

**Figure 3 cancers-14-05476-f003:**
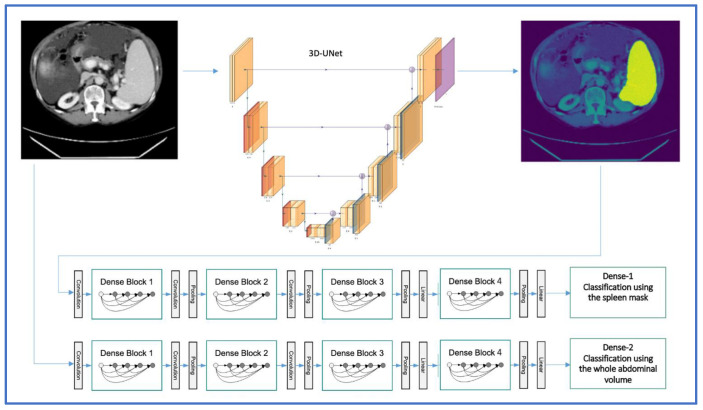
3D-UNET and DenseNet 121 architectures for the automated segmentation and classification of splenomegaly in the two-step approach (image adapted with permission from PlotNeuralNet [26], Copyright (c) 2018 HarisIqbal88. Available under the MIT license).

**Figure 4 cancers-14-05476-f004:**
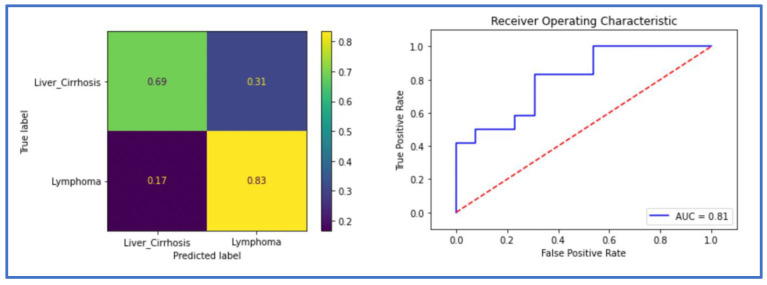
The receiver operating characteristic (ROC) curve and the confusion matrix of the classification model in the testing cohort using the spleen mask (Dense-Spl).

**Figure 5 cancers-14-05476-f005:**
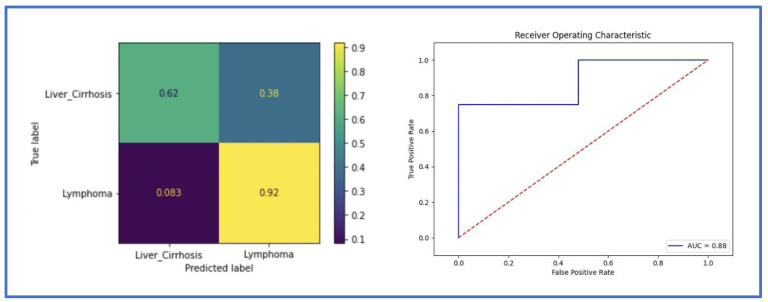
The receiver operating characteristic (ROC) curve and the confusion matrix of the classification model in the testing cohort using the whole abdominal volume (Dense-2).

**Figure 6 cancers-14-05476-f006:**
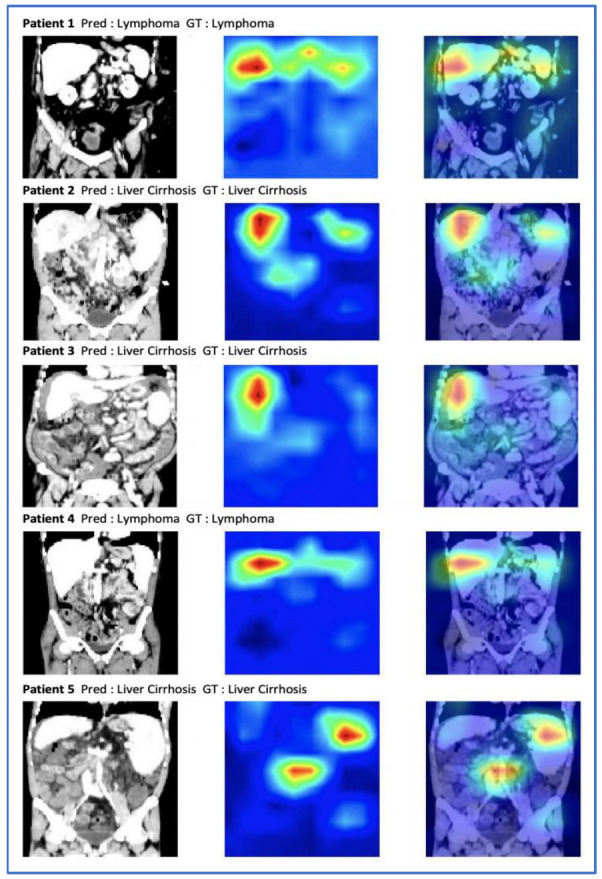
A selection of abdominal CT images with the ground truth (GT), the prediction (Pred), and the occlusion sensitivity maps superimposed on the input images. Red pixels correspond to a lower classification performance when occluded.

**Table 1 cancers-14-05476-t001:** Clinical factors of the training, validation, and testing cohorts. Unless otherwise indicated, data are expressed as the number of participants.

Clinical Factors	Training and Validation Cohort	Testing Cohort
Number of patients	124	25
Female	47 (38%)	12 (48%)
Age *	58.3 ± 14.7	56.2 ± 16
Cirrhosis with portal hypertension	64	13
Three most frequent lymphoma subtypes:		
DLBCL	13	6
FL	14	3
TCL	5	2
Other **	30	2

* Data are expressed as mean ±standard deviation; ** Other hematologic diseases include the remaining subtypes as described above.

**Table 2 cancers-14-05476-t002:** Metrics of the two models. AUC = area under the curve. ACC = accuracy. SEN = sensitivity. SPE = specificity.

DL Models	Training Cohort (*n* = 124)				Testing Cohort (*n* = 25)			
	AUC	ACC	SEN	SPE	AUC	ACC	SEN	SPE
Dense-Spl	0.84	0.84	0.77	0.91	0.81	0.76	0.69	0.83
ResNet-Spl	0.82	0.79	0.62	0.90	0.77	0.64	0.3	1
Dense-Abd	0.88	0.88	0.85	0.92	0.88	0.88	1	0.75
ResNet-Abd	0.86	0.84	0.76	0.91	0.80	0.80	0.69	0.91

## Data Availability

The data presented in this study dataset cannot be made publicly available for data protection reasons.

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
