# Peer review of "Two-Stage Deep Learning Model for Automated Segmentation and Classification of Splenomegaly"

_cancers, 2022, doi:10.3390/cancers14225476_

Round 1
Reviewer 1 Report
Authors classified splenic lesions based on clinical assumption not pathologic conformations. Therefore, they are confirming and original diagnostic bias, and not including other possibilities. They should have applied training and classifiers on one type of entities like mass forming lesions.
Reviewer 2 Report
The authors proposed a two-stage deep learning model to automatically segment the spleen and distinguish whether splenomegaly originates from lymphoma involvement or from cirrhotic portal hypertension. Authors concluded that automated segmentation of the spleen can be applied to detect and delineate splenomegaly, and the model can accurately differentiate between splenomegaly due to a malignant lymphoma and due to liver cirrhosis with portal hypertension.
The topic is interesting. However, I do have some major and minor concerns for the manuscript.
Major concerns:
1. Authors should add more technical details of the 3D-UNET and DenseNet architectures and data preparation;
2. In terms of the new model developed by authors, authors may consider how to demonstrate it is a novel method (e.g., compare test score with some other existing approaches)
Minor concerns:
1. Ln 143: authors may add citation for PlotNeuralNet;
2. Authors need to make clear labels for ground truth, prediction and occlusion sensitivity maps.
Reviewer 3 Report
Authors has proposed “Two-Stage Deep Learning Model for automated Segmentation and Classification of Splenomegaly ”. However, I have following comments.
· Abstract is too long, shorten it
· line 33, 34 fix the sentence, use of once is inappropriate
· line 42, 43, hanging sentence, fix it
· very weird of choices of words have been used. I suggest, do the proofreading of this manuscript.
· Literature of poorly mentioned, hardly any study has been discussed with properly way of explanation, no results, no outcomes has been mentioned in the literature.
· Author must consider mentioning following articles in the literature and make sure they elaborate them properly.
o https://ieeexplore.ieee.org/abstract/document/8533359/
o https://www.mdpi.com/2072-6694/14/8/2008
o http://137.158.155.148/handle/11427/35492
o https://www.mdpi.com/2076-3417/12/12/5990
o https://www.nature.com/articles/s41598-022-12410-2
o https://www.sciencedirect.com/science/article/abs/pii/S0925231222003149
o https://www.mdpi.com/2073-8994/13/11/1987
· Last para of introduction should precisely mention the contribution of the paper and followed by the reminder of the paper.
· Sample of the dataset (different CT images) must be presented in the manuscript in introduction.
· Dataset is relatively very small when it comes to different types of lymphoma subtypes
· Not much details about 3D- U-Net model has been provided (just a reference [13]). Authors much provide the detailed information about the model such as what are they modification they have done with the model, how they made this model to fix with this dataset, how segmentation has been performed?
· Another big question arises here, is that why authors have chosen U-Net model for segmentation , what were the means to choose that model? There are many models which show better results in segmentation Attention U-Net, TransUNet as mentioned in (https://www.mdpi.com/2076-3417/12/12/5990). I guess authors should provide the model selection process, provide Ablation Study in which they mentioned how U-Net is better than others and for those reasons they have chosen UNet over other models such as U-Net, V-Net, attention based (Attention) U-Net, TransUNet, Swin-UNet).
· Similar question about classification, out of blue author have decided to use Dense-Net. There is not any explanation provided why they have chosen Dense-Net architecture. Author must provide the Ablation study for choosing model for classification.
· Line 149, repeated “Results” heading
· Figure 5, no proper explanation provided, in Results you cant just summarize but need to provide detailed analysis what the model has done, how it has predicted and based on what it has predicted. I see nothing like that mentioned in the results. (section 3.2)
· Author show provide a qualitative as well as quantitative tables, comprising the proposed model finds with existing works found in same area. Without that it is difficult to identify the main contribution of this work.
· From line 209-248, it is not worth to mentioned it in the discussion whereas it can be pasted in the introduction or create a new section, related work and paste it there.
· No future direction provided in conclusion
·
Round 2
Reviewer 2 Report
The authors have made great revisions for the manuscript. My converns have been addressed.
Reviewer 3 Report
Authors have incorporated all the comments.